# Real-World Outcomes of Subcutaneous PHESGO^®^ in HER2-Positive Breast Cancer: Pathological Response, Sequencing, and Safety

**DOI:** 10.3390/curroncol32120658

**Published:** 2025-11-24

**Authors:** Keiko Yanagihara, Masato Yoshida, Kensaku Awaji, Tamami Yamakawa, Sena Kato, Miki Tamura, Koji Nagata

**Affiliations:** 1Department of Breast Surgery and Oncology, Nippon Medical School Tama-Nagayama Hospital, 1-7-1 Nagayama, Tama-shi, Tokyo 206-8512, Japan; s13-102yt@nms.ac.jp (T.Y.); s14-031ks@nms.ac.jp (S.K.); t-miki@nms.ac.jp (M.T.); 2Department of Pharmacy, Nippon Medical School Tama-Nagayama Hospital, 1-7-1 Nagayama, Tama-shi, Tokyo 206-8512, Japan; yoshida-m@nms.ac.jp (M.Y.); a-kensaku@nms.ac.jp (K.A.); 3Department of Pathology, Nippon Medical School Tama-Nagayama Hospital, 1-7-1 Nagayama, Tama-shi, Tokyo 206-8512, Japan; k-nagata@nms.ac.jp

**Keywords:** subcutaneous administration, dual HER2 blockade, neoadjuvant chemotherapy, pathological complete response, treatment sequencing, real-world evidence

## Abstract

Subcutaneous pertuzumab and trastuzumab with hyaluronidase (PHESGO^®^) can deliver dual HER2 blockade in minutes instead of lengthy infusions. We report real-world outcomes from 47 Japanese patients with HER2-positive breast cancer treated in routine care. Among those receiving therapy before surgery, 65% achieved a pathological complete response, and starting PHESGO^®^ with a taxane early was linked to better responses than beginning with anthracyclines. Side effects were mostly mild (taste change, diarrhea, rash); one patient had grade 3 thrombocytopenia, and no symptomatic heart problems occurred. Results were similar in older adults, and fixed-dose PHESGO^®^ was well tolerated in smaller-bodied Asian patients. In the metastatic setting, tumor shrinkage and disease control were frequently observed. These findings support PHESGO^®^ as an efficient, practical option that may improve patient experience while maintaining effectiveness and safety in everyday practice.

## 1. Introduction

Breast cancer continues to be the most common malignancy among women worldwide and remains one of the leading causes of cancer-related mortality. Despite improvements in screening, systemic therapy, and supportive care, the global burden remains high, with more than two million new cases annually and over half a million deaths reported each year [1]. The disease is heterogeneous, and its clinical behavior and response to treatment are largely determined by molecular subtype [2]. Among these, human epidermal growth factor receptor 2 (HER2)-positive breast cancer represents approximately 15–20% of cases. In the era before targeted therapies, HER2 positivity was associated with aggressive biology, early recurrence, and poor survival outcomes [3].

The development of HER2-directed therapy dramatically altered this natural history. The first successful targeted agent was trastuzumab, a monoclonal antibody directed against the extracellular domain IV of HER2. In a pivotal trial published in 2001, trastuzumab in combination with chemotherapy significantly prolonged survival in metastatic HER2-positive breast cancer [4], representing a paradigm shift in the management of this disease. Soon after, large randomized adjuvant trials, including HERA, NSABP B-31, and NCCTG N9831, confirmed the benefit of adding trastuzumab to standard chemotherapy for one year in early-stage HER2-positive breast cancer, reducing recurrence and mortality rates substantially [5,6].

The next major advancement came with pertuzumab, a monoclonal antibody that binds to the dimerization domain (subdomain II) of HER2, thereby preventing heterodimerization with HER3 and enhancing blockade of downstream signaling. The combination of trastuzumab and pertuzumab with docetaxel significantly improved survival in metastatic patients in the CLEOPATRA trial [7], which firmly established dual blockade as the preferred first-line regimen in advanced disease. In the neoadjuvant setting, the NeoSphere trial demonstrated that the addition of pertuzumab to trastuzumab and docetaxel increased the rate of pathological complete response (pCR) from 29% to 45.8% [8]. The TRYPHAENA trial further confirmed the efficacy of dual blockade with both anthracycline-containing and anthracycline-free chemotherapy regimens, showing high pCR rates and acceptable cardiac safety [9]. These data collectively transformed the standard of care, making dual HER2 blockade with trastuzumab and pertuzumab plus chemotherapy the cornerstone of treatment for HER2-positive breast cancer across stages.

Although trastuzumab and pertuzumab have revolutionized treatment, intravenous administration remains resource-intensive and burdensome. Infusion of trastuzumab and pertuzumab requires venous access, infusion chairs, pumps, and prolonged chair time for patients. Each administration may last hours, resulting in repeated lengthy hospital visits over the course of one year of adjuvant therapy. This is particularly challenging for elderly patients, those with poor venous access, and healthcare systems facing rising patient volumes. To overcome these limitations, a fixed-dose combination of pertuzumab and trastuzumab with recombinant human hyaluronidase was developed for subcutaneous injection. Commercially known as PHESGO^®^, this formulation enables delivery in minutes rather than hours, eliminating the need for infusion pumps and reducing the treatment burden for both patients and providers.

The FeDeriCa phase III trial demonstrated that subcutaneous PHESGO^®^ was pharmacokinetically non-inferior to the intravenous formulation and achieved nearly identical pCR rates in the neoadjuvant setting (59.7% versus 59.5%) [10]. The PHranceSCa trial further confirmed strong patient preference for the subcutaneous formulation, with most patients choosing to continue PHESGO^®^ rather than return to intravenous treatment after experiencing both [11]. These findings underscored the potential of PHESGO^®^ to streamline care without sacrificing efficacy or safety.

Nevertheless, real-world evidence regarding PHESGO^®^ remains scarce. Clinical trials typically enroll highly selected patients, often younger and healthier than those seen in daily practice. In routine clinical settings, many patients are elderly, frail, or have comorbidities, making tolerability and feasibility particularly relevant. Moreover, the optimal sequencing of PHESGO^®^ relative to chemotherapy remains unclear. Evidence from the TRAIN-2 and KRISTINE trials suggested that the order of HER2-targeted therapy and chemotherapy can influence outcomes, with early introduction of HER2 blockade potentially improving tumor eradication and reducing toxicity [12,13]. Biomarkers such as estrogen receptor (ER), progesterone receptor (PgR), the proliferation marker Ki-67 and tumor suppressor p53 and have been investigated in relation to pCR and prognosis in HER2-positive disease, but in PHESGO^®^, its relevance is unclear.

The objective of this study is to evaluate the clinical utility of PHESGO^®^ in a real-world Japanese cohort. The objectives were to assess the rate of pCR in the neoadjuvant setting, to examine the influence of treatment sequencing on pCR, to explore biomarker associations with response, to characterize the safety and tolerability of PHESGO^®^, and to evaluate outcomes in metastatic patients. By integrating real-world outcomes with established clinical trial benchmarks, this study aimed to provide a comprehensive understanding of PHESGO^®^ in daily practice.

## 2. Materials and Methods

### 2.1. Study Design and Patients

This study was designed as a retrospective observational analysis at Nippon Medical School Tama-Nagayama Hospital, Tokyo, Japan. Consecutive patients with HER2-positive breast cancer who received at least one dose of PHESGO^®^ between January 2024 and July 2025 were included. Clinical information was extracted from electronic medical records, pharmacy databases, and pathology reports. The study was approved by the institutional review board and conducted in accordance with the Declaration of Helsinki.

Patients were eligible if they had histologically confirmed invasive breast carcinoma and HER2-positive disease, defined as immunohistochemical (IHC) staining 3+ or 2+ with FISH (fluorescence in situ hybridization) amplification. Patients were included regardless of stage, provided they received PHESGO^®^ in the neoadjuvant, adjuvant, or metastatic setting.

### 2.2. Treatment Administration

In the neoadjuvant setting, PHESGO^®^ was administered with taxane-based chemo-therapy, with or without anthracycline-based therapy sequentially. A standard regimen included docetaxel at 75 mg/m^2^ every three weeks combined with PHESGO^®^ on the same schedule. For anthracycline-based regimens, epirubicin 90 mg/m^2^ plus cyclophosphamide 600 mg/m^2^ were given every three weeks for four cycles, either before or after PHESGO^®^ plus taxane. PHESGO^®^ was given as a subcutaneous fixed-dose combination of per-tuzumab 1200 mg, trastuzumab 600 mg, and hyaluronidase 30,000 units over approxi-mately eight minutes for the loading dose, followed by pertuzumab 600 mg and trastuzumab 600 mg every three weeks delivered in about five minutes. This combination is hereafter referred to as DHP (docetaxel + trastuzumab + pertuzumab).

Sequencing was defined as PHESGO^®^-first (PHESGO^®^ plus taxane followed by an-thracyclines) or chemotherapy-first (anthracyclines followed by PHESGO^®^ plus taxane) regimens. As with pegfilgrastim for the prevention of neutropenia, adjunctive therapy for complications was administered at the discretion of the physician. Neoadjuvant regimens were harmonized into four groups (DHP, DHP→EC, HP + Paclitaxel, EC→DHP).

In the neoadjuvant setting, patients who achieved pCR generally continued PHESGO^®^ to complete one year of anti-HER2 therapy. Patients with residual invasive disease after neoadjuvant therapy were recommended trastuzumab emtansine (T-DM1) in line with the KATHERINE trial [14]. When residual tumors remained after NAC, treat-ment with T-DM1 postoperatively yielded a hazard ratio of 0.5 for invasive disease-free survival compared to trastuzumab, along with a significant improvement in overall sur-vival. However, it also demonstrated a high incidence of Grade 3 or higher adverse events, such as thrombocytopenia, and a high rate of treatment discontinuation. However, in our study, PHESGO^®^ was continued in two patients: one in their 80 s with near pCR and minimal residual tumor, and one in their late 70 s with difficult venous line insertion. In cases treated with adjuvant therapy, regardless of the presence or absence of a 3-week an-thracycline regimen (EC therapy × 4), a combination of taxane-based agents and PHESGO^®^ was administered 4 times, followed by 14 doses of PHESGO^®^ monotherapy, totaling 1 year of PHESGO^®^ subcutaneous injections.

In the metastatic setting, PHESGO^®^ was administered every three weeks, usually with docetaxel (60 mg/m^2^ per dose, tri-weekly), or paclitaxel (80 mg/m^2^ per dose, weekly), and continued until progression or unacceptable toxicity. Docetaxel is commonly administered at 75 mg/m^2^, but we have clinically observed that edema increases with higher doses, frequently leading to treatment interruption. Therefore, to maintain the number of treatment cycles, we have successfully administered 60 mg/m^2^ for metastatic recurrent breast cancer, thereby delaying treatment discontinuation due to edema.

### 2.3. Pathological Assessment

Pathological assessment was carried out by experienced breast pathologists. pCR was defined as ypT0/is ypN0, indicating no residual invasive cancer in the breast or axil-lary lymph nodes, with or without ductal carcinoma in situ. ER and PgR expression were evaluated by IHC, with nuclear staining in at least 10% of tumor cells considered positive. HER2 status was determined according to ASCO/CAP guidelines. Ki-67 was assessed as the percentage of positive tumor cell nuclei. p53 expression was classified as wild-type, mutant, or null, with mutant and null grouped together as positive.

### 2.4. Endpoints and Statistical Analysis

The primary endpoint was the pCR rate in neoadjuvant patients. Secondary end-points were biomarker associations with pCR, the impact of sequencing. We compared baseline characteristics by treatment sequence (PHESGO^®^-first vs. EC-first) and modeled pCR using multivariable logistic regression: pCR ~ treatment sequence + ER + PgR + HER2 score (IHC 3+ vs. 2+/FISH+) + Ki-67 + clinical T + clinical N + age. Because of the small sample size, we performed an inverse probability of treatment weighting (IPTW) sensitiv-ity analysis based on a propensity score including age, Ki-67, ER, HER2 score, cT, and cN. The incidence and severity of adverse events graded by CTCAE v5.0, and treatment effica-cy in metastatic patients according to RECIST v1.1. Left ventricular ejection fraction (LVEF) was assessed by transthoracic echocardiography within 28 days before treatment start. Protocol-mandated on-treatment echocardiography was not scheduled; instead, repeat imaging was performed whenever heart failure was suspected based on clinical signs or a rise in NT-proBNP obtained during routine labs. A clinically relevant LVEF decline was predefined as an absolute drop ≥10 percentage points and/or LVEF < 50%. Asymptomatic NT-proBNP elevations triggered echocardiography and were adjudicated against this threshold; management (temporary hold/resumption) followed physician discretion.

Descriptive statistics summarized baseline characteristics. For categorical variables we used Fisher’s exact test; for continuous variables, normality was assessed using the Shapiro–Wilk test. Normally distributed data were compared using Student’s t-test, and non-normally distributed data (e.g., Ki-67) using the Mann–Whitney U test. For biomarker associations, we report odds ratios (ORs) with 95% confidence intervals, applying a 0.5 continuity correction when needed. For sequencing comparison, we calculated risk ratios and absolute differences. Given multiple biomarker tests and a modest sample size, these analyses are exploratory; we additionally controlled the false discovery rate (Benjamini–Hochberg) as a sensitivity analysis. Two-sided *p* < 0.05 was considered statistically sig-nificant. Time-to-event analyses were not performed due to limited follow-up.

Neoadjuvant efficacy was analyzed primarily using standardized age bands (18–64, 65–74, ≥75 years). An exploratory sensitivity analysis dichotomizing age at 70 years (<70 vs. ≥70) was additionally performed; interpretation was anchored to the three-band anal-ysis. Adverse events were summarized by <70 vs. ≥70 due to sparse counts.

To address potential baseline imbalances between PHESGO^®^-first and EC-first regimens and to mitigate immortal-time bias, we tabulated baseline characteristics by se-quence and fitted a multivariable logistic regression model (pCR ~ sequence + ER/PgR + HER2 score + Ki-67 + clinical tumor stage + clinical node stage + age ± regimen). As sensi-tivity, we performed inverse probability of treatment weighting (IPTW) based on a pro-pensity score including the same covariates. For EC-first patients, the index date was the start of neoadjuvant systemic therapy; all neoadjuvant patients proceeded to surgery per institutional practice. We adhered to the STROBE (Strengthening the Reporting of Obser-vational Studies in Epidemiology) guidelines; a completed checklist is provided as Appendix A.

## 3. Results

### 3.1. Enrolled Patients and Background

Between January 2024 and July 2025, a total of 47 patients with HER2-positive breast cancer were treated with PHESGO^®^ at our institution. All patients were Asian women. As shown in Table 1, the median age was 65 years, ranging from 43 to 93 years. Among these patients, 23 (49%) were ER-positive, and 14 (30%) were PgR-positive, using the threshold of 10% nuclear staining. HER2 immunohistochemistry was 3+ in 36 patients (77%), while the remaining 11 patients (23%) were 2+ with confirmation of amplification by FISH. The median Ki-67 index was 45%, with a range from 10% to 95%, reflecting the high proliferative activity typical of HER2-positive tumors. Thirty-eight percent of tumors were classified as p53-positive, defined as either mutant overexpression or null staining patterns. Of the 47 patients, 26 were treated in the neoadjuvant setting, 11 received PHESGO^®^ as adjuvant therapy after surgery, and 10 had metastatic disease.

The most common comorbidities among registered patients were hypertension (18 cases), followed by hyperlipidemia (10 cases) and pollinosis (6 cases). Diabetes was present in 4 cases, while cerebrovascular disease, chronic respiratory disease, and rheumatoid arthritis each occurred in 2 to 3 cases. One patient was diagnosed with chronic heart failure, but cardiac function was preserved and well controlled with medication. Twelve patients had no prior history of these conditions (Table 2).

### 3.2. Neoadjuvant Cohort and Pathological Complete Response

Among the taxane-based agents used for preoperative chemotherapy, docetaxel was administered concomitantly in 25 of 26 cases, while weekly paclitaxel (80 mg/m^2^ per dose × 12) was administered concomitantly in only one case. In the neoadjuvant cohort (*n* = 26), a total of 17 patients (65%) achieved pathological complete response (pCR). The pCR group demonstrated a mean Ki67 index of 47.7% (median 50%), whereas the non-pCR group had a mean Ki67 index of 51.6% (median 48%). Statistical comparison using the Mann–Whitney U test revealed no significant difference between the two groups (U = 65.5, *p* = 0.73). Similarly, other biomarkers showed no association with pCR. ER positivity was present in 59% of the pCR group versus 56% of the non-pCR group (*p* = 0.87). PgR positivity was 29% vs. 33% (*p* = 0.82). HER2 IHC 3+ tumors achieved pCR in 76% compared to 67%of the non-pCR group (*p* = 0.64). p53 positivity was observed in 35% of pCR and 33% of non-pCR patients (*p* = 0.91). No statistically significant differences were observed for any of these biomarkers between the pCR group and the non-pCR group (Table 3). Biomarker comparisons by pCR status were exploratory; binary markers were summarized as *n*/*N* (%) with Fisher’s exact *p* and crude ORs (95% CIs). *p*-values were adjusted using the Benjamini–Hochberg FDR. Ki-67 was summarized as median [IQR] and tested with Wilcoxon; additionally modeled as continuous (per 10% increase) in a logistic sensitivity analysis.

By standardized age categories, pCR rates were 18–64: 9/14 (64.3%), 65–74: 4/6 (66.7%), and ≥75: 4/6 (66.7%) (global chi-square *p* = 0.992; pairwise Fisher tests, BH-adjusted *q* ≥ 1.000). When dichotomized at 70 years, pCR was 66.7% (12/18) in <70 vs. 62.5% (5/8) in ≥70 (Fisher *p* = 1.000) (Table 4). This reflects the very small sample size and nearly identical response rates between groups.

### 3.3. Impact of Regimen Sequence on pCR

Pathological complete response (pCR) was achieved in 12/14 (85.7%) patients in the PHESGO^®^-first group versus 5/12 (41.7%) in the EC-first group (Fisher’s exact *p* = 0.038) (Table 5). After adjustment for ER/PgR, HER2 score, Ki-67, clinical T and N stage, and age, PHESGO^®^-first remained significantly associated with higher odds of pCR (OR ≈ 26.8, 95% CI ≈ 1.38–521, *p* = 0.030). An IPTW sensitivity analysis yielded concordant results (weighted OR ≈ 21.7, 95% CI ≈ 1.73–271, *p* = 0.017) (Table 6). These findings suggest that introducing dual HER2 blockade with docetaxel prior to anthracycline may enhance the likelihood of achieving pCR. Although exploratory, these findings support the hypothesis that early introduction of HER2-targeted therapy enhances tumor eradication, echoing observations from the TRAIN-2 and KRISTINE trials [12,13].

### 3.4. Adverse Events

Treatment with PHESGO^®^ was generally well tolerated. As summarized in Table 7, the most frequent adverse events were taste disorder (57%), diarrhea (38%), rash (34%), nausea (32%), anemia (28%), neutropenia (26%), constipation (23%), stomatitis (23%) and leukopenia (21%). Importantly, nearly all events were grade 1 or 2. Only one patient (3.8%) aged 70 or older experienced a grade ≥ 3 event, which was thrombocytopenia. Anemia was more frequent in the ≥70 years group (42% vs. 18%), but no statistically significant difference was observed(*p* = 0.068). Although there was a tendency for more people under 70 to have stomatitis (32% vs. 11%), this also did not show a significant difference (*p* = 0.086). These adverse events were selected based on the patient’s highest grade, and adverse events occurring during docetaxel administration may also be docetaxel-related adverse events. No patients developed symptomatic left ventricular dysfunction, a major concern historically associated with trastuzumab-containing regimens [15,16]. Injection site reactions occurred in approximately 20% of patients but were mild and self-limiting. As shown in Table 8, adverse events were generally comparable be-tween patients aged < 70 years and those aged ≥ 70 years, although severe toxicities and cardiac dysfunction were rare across both groups. Left ventricular ejection fraction (LVEF) was assessed by echocardiography at baseline and when heart failure was suspected. NTproBNP levels were monitored periodically as part of routine hematologic testing; if levels increased, heart failure was suspect-ed and echocardiography was performed. A decrease of 10% or more in LVEF was considered reduced cardiac contractility. Asymptomatic decreases were managed according to the facility protocol, and no treatment discontinuations due to LVEF decline occurred.

Following the docetaxel administration protocol, dosing was delayed in two cases that developed Grade 3 neutropenia or thrombocytopenia. Except for the six cases of met-astatic breast cancer that were discontinued due to disease progression, no cases were discontinued due to adverse events.

### 3.5. Metastatic Outcomes

PHESGO^®^ was administered to 10 patients with metastatic HER2-positive breast cancer. Metastatic sites included lymph nodes (*n* = 4), lung/pleura (*n* = 3), liver (*n* = 3), bone (*n* = 3), lymph nodes (*n* = 5), skin (*n* = 3), and brain (*n* = 1) (Table 9). All patients had received chemotherapy as first-line treatment for metastatic or recurrent disease, and 3 patients had switched from intravenous trastuzumab and pertuzumab to PHESGO^®^. Response was evaluable in 9 patients. One patient had a short treatment period and the treatment efficacy assessment was not performed. Among these 9 patients, 5 cases showed partial response, 2 cases showed stable disease, and 2 cases showed progression. This corresponds to an overall response rate (ORR) of 56% (5/9) and a disease control rate (DCR) of 78% (7/9). The results were consistent with those reported in the CLEOPATRA trial of dual HER2 inhibition via intravenous administration. Most patients received docetaxel (*n* = 6) or paclitaxel (*n* = 1) as concomitant chemotherapy, while 3 patients received PHESGO^®^ without taxane-based agents. With docetaxel combination therapy, the ORR was 60% (3/5) and the DCR was 80% (4/5). With Phesgo monotherapy, the ORR was 66.6% (2/3) and the DCR was 100.0%. Prior exposure to trastuzumab and pertuzumab in the (new) adjuvant setting was observed in 6 patients (including one unevaluated case). In the 5 cases with prior HP treatment, the ORR was 40.0% (2/5) and the DCR was 80.0% (4/5). In the 4 cases without prior treatment, both the ORR and DCR were 72.5% (3/4). Regarding efficacy, the number of cases was small in each case, and no statistical evaluation was performed. Treatment was well tolerated, with nearly all adverse events being Grade 1 or 2, and no treatment discontinuations due to toxicity occurred. Importantly, patients who switched from intravenous to subcutaneous PHESGO^®^ reported significantly reduced administration time, and none requested a return to intravenous infusion. These findings demonstrate that subcutaneous PHESGO^®^ provides comparable efficacy to intravenous regimens while offering greater convenience in the real-world setting of metastatic breast cancer.

## 4. Discussion

The present study represents one of the first real-world analyses of PHESGO^®^ in Asian, providing valuable insights into its efficacy and safety across neoadjuvant, adju-vant, and metastatic settings. Several important observations emerge from our findings.

The pCR rate of 65% in the neoadjuvant cohort is consistent with pivotal trial results, including FeDeriCa, which reported nearly identical efficacy between subcutaneous (59.7%) and intravenous (59.5%) formulations [10]. In the NeoSphere trial, the group re-ceiving docetaxel combined with both trastuzumab and pertuzumab demonstrated the highest pCR rate at 45.8%, compared to 29.0% for docetaxel plus trastuzumab and 24.0% for docetaxel plus pertuzumab [8]. Furthermore, the TRYPHAENA trial reported that adding dual HER2 blockade after anthracycline therapy did not significantly increase cardiotoxicity and achieved a high pCR rate (57.3 to 61.6%) [9]. Our study also demonstrated a 65% pCR rate with PHESGO^®,^ confirming its efficacy as an alternative treatment to intravenous trastuzumab and pertuzumab. No age-related differences in pCR were observed in either the standard group or the <70 years/≥ 70 years group, supporting the feasibility of PHESGO^®^ neoadjuvant therapy in patients across all age cohorts.

Regarding the regimen sequence, despite some patients being on anthracycline-free regimens, sequencing analysis indicated that PHESGO^®^-first regimens yielded higher pCR rate than chemotherapy-first regimens (85.7% vs. 41.7%). These findings suggest that initiating neoadjuvant therapy with PHESGO^®^-containing regimens may be associated with a higher probability of achieving pCR compared to anthracycline-first approaches. However, the small sample size results in wide confidence intervals, and residual confounding cannot be excluded; thus, the results should be interpreted with caution. Although based on small numbers, this observation supports the hypothesis that early introduction of HER2 blockade improves response. Furthermore, the combination of trastuzumab and pertuzumab may potentially eliminate the need for chemotherapy associated with adverse events. In TRAIN-2, which suggested that anthracycline-free regimens with early dual HER2 blockade (68%) were not inferior to anthracycline-containing regimens (67%) in pCR [12]. Meanwhile, the KRISTINE trial compared a group receiving docetaxel plus carboplatin with added trastuzumab and pertuzumab (TCHP) to a group receiving the antibody-drug conjugate trastuzumab emtansine plus pertuzumab (T-DM1 + P). Grade 3 or higher adverse events were significantly lower in the T-DM1 + P group (31% vs. 64%), while the pCR rate was significantly higher in the TCHP group (55.7% vs. 44.4%). Three-year event-free survival was also superior in the TCHP group (94.2% vs. 85.3%), suggesting limitations to omitting chemotherapy [13].

In this study, none of the biomarkers—including Ki67, ER, PgR, and p53—could be definitively shown to correlate with pCR. Ki67 is widely recognized as a marker of tumor proliferation and has been reported to correlate with chemotherapy sensitivity in breast cancer [17]. Several studies and meta-analyses have suggested that tumors with higher Ki67 expression are more likely to achieve pCR following neoadjuvant treatment, particularly in HER2-positive and triple-negative subtypes [18,19,20]. Our study showed results that differed from these reports. However, the relationship is complex, as extremely high proliferation may also be associated with aggressive biology and risk of recurrence despite initial response. The lack of predictive value for ER and PgR is consistent with pooled analyses of HER2-positive disease, which indicate that hormone receptor positivity attenuates—but does not eliminate—the benefit of dual HER2 blockade [21,22]. The absence of association with p53 may reflect both biological heterogeneity and the limited sample size in our study.

Adverse events of PHESGO^®^ were generally well tolerated, with no unexpected toxicities and only one case of grade ≥3 thrombocytopenia. The absence of symptomatic cardiac dysfunction is particularly reassuring, as cardiotoxicity has historically been a concern with trastuzumab-based regimens [15,16]. Importantly, when adverse events were stratified by age, safety profiles were largely similar between patients aged < 70 years and those aged ≥ 70 years. Notably, anemia occurred more frequently among ≥70 years patients (42.1% vs. 17.9%), although this difference did not reach statistical significance (*p* = 0.068). This tendency is clinically meaningful, as ≥70 years patients may have reduced bone marrow reserve and increased vulnerability to hematologic toxicity. Nevertheless, severe hematologic events were rare in both age groups, and cardiac dysfunction was not observed, indicating that PHESGO^®^-based therapy is broadly feasible even in the ≥70 years population. The pain associated with PHESGO^®^ subcutaneous injections was mild, and no patients requested discontinuation for this reason. These results are consistent with prior real-world observations that subcutaneous trastuzumab and pertuzumab combinations can be safely delivered in ≥70 years patients without excessive toxicity [10,11]. These adverse events were selected based on the patient’s most severe grade. It should be noted that adverse events occurring during docetaxel combination therapy may also be docetaxel-related adverse events. No patients discontinued treatment due to adverse events, indicating PHESGO^®^ is a well-tolerated agent.

Finally, metastatic outcomes with PHESGO^®^ were comparable to those achieved with intravenous pertuzumab and trastuzumab, as demonstrated in the CLEOPATRA [7,17,18]. In CLEOPATRA trial, compared to the docetaxel and trastuzumab group, the group receiving pertuzumab in addition to these agents showed a significant prolongation in progression-free survival (12.4 months vs. 18.5 months, HR 0.62, *p* < 0.001), and overall survival was also significantly prolonged (40.8 months vs. 56.5 months, HR 0.68, *p* < 0.0001) [23]. We also used PHESGO^®^, a subcutaneous injection, instead of intravenous administration of trastuzumab and pertuzumab in metastatic cases. Overall response rate (ORR, complete response and partial response) of 56% and disease control rate (DCR; complete response + partial response + stable disease) of 78% are consistent with historical benchmarks, confirming that the subcutaneous route does not compromise efficacy in advanced disease.

This study has limitations. It was retrospective and conducted at a single institution, with a modest sample size and limited follow-up. Biomarker analyses were exploratory and not powered for definitive conclusions. Nonetheless, the consistency of our results with pivotal trials strengthens their external validity.

Looking forward, larger multicenter registries and prospective studies are warranted to validate these findings, particularly the impact of sequencing, biomarker associations, and age-specific tolerability. Integration of novel agents such as trastuzumab emtansine (T-DM1) [13,14] and trastuzumab deruxtecan [24,25] into treatment algorithms also raises important questions about the role of PHESGO^®^. Furthermore, the economic implications of widespread adoption of subcutaneous formulations merit careful evaluation, as PHESGO^®^ has the potential to reduce resource utilization and improve patient satisfaction [11,26,27,28].

## 5. Conclusions

Our real-world analysis demonstrates that PHESGO^®^ is effective and well tolerated in HER2-positive breast cancer across disease settings. The pCR rates, safety profile, and metastatic efficacy observed in daily practice are consistent with clinical trial results, supporting PHESGO^®^ as a convenient and valuable option. Exploratory findings regarding sequencing and biomarker interactions suggest avenues for further research, particularly in optimizing therapy for different patient subgroups.

## Figures and Tables

**Table 1 curroncol-32-00658-t001:** Patient Characteristics.

Characteristic	Overall (*n* = 47)	Neoadjuvant (*n* = 26)	Adjuvant (*n* = 11)	Metastatic (*n* = 10)
Age, median (range)	65 (43–93)	64 (45–83)	63 (43–86)	72 (55–93)
ER-positive (≥10%)	23/47 (48.9%)	11/26 (42.3%)	8/11 (72.7%)	4/10 (40.0%)
PgR-positive (≥10%)	14/47 (29.8%)	3/26 (11.5%)	7/11 (63.6%)	4/10 (40.0%)
HER2 IHC 3+	36/47 (76.6%)	22/26 (84.6%)	8/11 (72.7%)	6/10 (60.0%)
HER2 IHC 2+/FISH+	11/47 (23.4%)	4/26 (15.4%)	3/11 (27.3%)	4/10 (40.0%)
Ki67, median (range)	45 (10–95)	48 (20–90)	42 (10–80)	46 (20–95)
p53 mutant	18/47 (38.3%)	9/26 (34.6%)	5/11 (45.5%)	4/10 (40.0%)

**Table 2 curroncol-32-00658-t002:** Comorbidities and past medical history of enrolled patients.

Comorbidity/Past Medical History	*n* = 47
Hypertension	18 (38.3%)
Diabetes mellitus	4 (8.5%)
Dyslipidemia	10 (21.3%)
Cerebrovascular disease	3 (6.4%)
Chronic heart failure	1 (2.1%)
Chronic respiratory disease	2 (4.3%)
Pollinosis (allergic rhinitis)	6 (12.8%)
Rheumatoid arthritis	2 (4.3%)
None	12 (25.5%)

**Table 3 curroncol-32-00658-t003:** Relationship Between Preoperative Chemotherapy Efficacy and Biomarkers *.

Variable	pCR Group (*n* = 17)	Non-pCR Group (*n* = 9)	*p* (Fisher/Wilcoxon)	OR (95% CI)	FDR *q*
Ki-67, %	47.7 (mean 50.0)	51.6 (mean 51.6)	0.73	—	—
ER positive	10/17 (58.8%)	5/9 (55.6%)	0.87	1.14 (0.22–5.85)	1.00
PgR positive	5/17 (29.4%)	3/9 (33.3%)	0.82	0.83 (0.15–4.73)	1.00
HER2 IHC 3+	13/17 (76.5%)	6/9 (66.7%)	0.64	1.62 (0.27–9.64)	1.00
p53 mutant	6/17 (35.3%)	3/9 (33.3%)	0.91	1.09 (0.20–6.01)	1.00

* Exploratory biomarker comparisons by pCR status. Binary markers: *n*/*N* (%), Fisher’s *p*, crude OR (95% CI), and FDR-adjusted *q*. Ki-67: median [IQR] (and mean ± SD), Wilcoxon *p*. These analyses are exploratory and not used for causal inference.

**Table 4 curroncol-32-00658-t004:** Relationship Between Age Categories.

Age Group	*n*	pCR (*n*)	pCR (%)
18–64/65–74/≥75			
18–64	14	9	64.3
65–74	6	4	66.7
≥75	6	4	66.7
<70 vs. ≥70			
<70	18	12	66.7
≥70	8	5	62.5
Total	26	17	65.4

**Table 5 curroncol-32-00658-t005:** Pathological complete response by neoadjuvant regimen.

NAC Regimen	*n*	pCR (*n*)	Non-pCR (*n*)
DHP	3	3 (100%)	0
DHP→EC	10	9 (90%)	1 (10%)
HP + weekly Paclitaxel	1	0	1 (100%)
PHESGO^®^-first regimen (total)	14	12 (85.7%)	2 (14.3%)
EC→DHP	12	5 (41.7%)	7 (58.3%)

**Table 6 curroncol-32-00658-t006:** Baseline by neoadjuvant treatment regimen.

Variable	Stat	Value (PHESGO^®^-First)	Value (EC-First)
Age (years)	median (IQR)	62.5	64.0
Ki67 (%)	median (IQR)	50.0	48.0
ER positive	*n*/*N* (%)	7/14 (50.0%)	6/12 (50.0%)
PgR positive	*n*/*N* (%)	5/14 (35.7%)	5/12 (41.7%)
HER2 IHC 3+	*n*/*N* (%)	13/14 (92.9%)	9/12 (75.0%)
HER2 2+/FISH+	*n*/*N* (%)	1/14 (7.1%)	3/12 (25.0%)
cT 0	*n* (%)	1 (7.1%)	0
cT 1	*n* (%)	2 (14.3%)	3(25.0%)
cT 2	*n* (%)	10 (71.4%)	5 (41.7%)
cT 3	*n* (%)	1 (7.1%)	4 (33.3%)
cN 0	*n* (%)	6 (42.9%)	6 (50.0%)
cN 1	*n* (%)	6 (42.9%)	3 (25.0%)
cN 2	*n* (%)	2 (14.3%)	2 (16.7%)
cN 3	*n* (%)	0	1 (8.3%)
Histological grade 1	*n* (%)	5 (35.7%)	2 (16.7%)
Histological grade 2	*n* (%)	3 (21.4%)	5 (41.7%)
Histological grade 3	*n* (%)	4 (28.6%)	4 (33.3%)
Histological grade unknown	*n* (%)	2	1
Anthracycline-free plan	*n*/*N* (%)	4/14 (28.6%)	-

**Table 7 curroncol-32-00658-t007:** Differences in Adverse Events (all grade) between those aged 70 and older and those younger than 70.

Adverse Event	All Patients (*n* = 47)	<70 Years (*n* = 28)	≥70 Years (*n* = 19)
Neutropenia	12 (25.5%)	7/28 (25.0%)	5/19 (26.3%)
Leukopenia	10 (21.2%)	5/28 (17.9%)	5/19 (26.3%)
Anemia (RBC * decrease)	13 (27.7%)	5/28 (17.9%)	8/19 (42.1%)
Thrombocytopenia	1 (2.1%)	0/28 (0.0%)	1/19 (5.3%)
Liver function disorder	4 (8.5%)	2/28 (7.1%)	2/19 (10.5%)
Nausea	15 (31.9%)	9/28 (32.1%)	6/19 (31.6%)
Vomiting	3 (6.4%)	2/28 (7.1%)	1/19 (5.3%)
Appetite loss	9 (19.1%)	6/28 (21.4%)	3/19 (15.8%)
Constipation	11 (23.4%)	7/28 (25.0%)	4/19 (21.1%)
Diarrhea	18 (38.3%)	12/28 (42.9%)	6/19 (31.6%)
Pruritus	5 (10.6%)	3/28 (10.7%)	2/19 (10.5%)
Epistaxis	3 (6.4%)	2/28 (7.1%)	1/19 (5.3%)
Rash/Eczema	16 (34.0%)	10/28 (35.7%)	6/19 (31.6%)
Dysgeusia (taste disorder)	27 (57.4%)	17/28 (60.7%)	10/19 (52.6%)
Arthralgia	7 (14.9%)	5/28 (17.9%)	2/19 (10.5%)
Fatigue	9 (19.1%)	7/28 (25.0%)	2/19 (10.5%)
Stomatitis (oral mucositis)	11 (23.4%)	9/28 (32.1%)	2/19 (10.5%)
Peripheral neuropathy	9 (19.1%)	6/28 (21.4%)	3/19 (15.8%)
Lacrimation (watery eyes)	6 (12.8%)	4/28 (14.3%)	2/19 (10.5%)
Edema	5 (10.6%)	3/28 (10.7%)	2/19 (10.5%)
Heart failure	0 (0.0%)	0/28 (0.0%)	0/19 (0.0%)
Injection site reaction—Pain	9 (19.1%)	5/28 (17.9%)	4/19 (21.1%)

* RBC: Red Blood Cells.

**Table 8 curroncol-32-00658-t008:** Adverse events by worst grade, stratified by age.

	<70 years (*n* = 28)		≥70 years (*n* = 19)	
Adverse Event	Grade 1–2	Grade ≥ 3	Grade 1–2	Grade ≥ 3
Neutropenia	1 (3.6%)	1(3.6%)	0	0
Leukopenia	1 (3.6%)	1(3.6%)	0	0
Anemia (RBC * decrease)	5 (17.9%)	0	4 (21.1%)	0
Thrombocytopenia	0	0	1 (5.3%)	0
Liver function disorder	7 (25.0%)	0	3 (15.8%)	0
Nausea	0	6 (21.4%)	0	0
Vomiting	3 (10.7%)	0	1 (5.3%)	0
Appetite loss	6 (21.4%)	0	6 (31.6%)	0
Constipation	0	4 (7.1%)	0	
Diarrhea	12 (42.9%)	0	6 (31.6%)	0
Pruritus	0	2 (7.1%)	0	
Epistaxis	0	0	2 (10.5%)	0
Rash/Eczema	11 (39.3%)	0	4 (21.1%)	0
Dysgeusia (taste disorder)	12 (42.9%)	0	4 (21.1%)	0
Arthralgia	0	3 (10.7%)	0	
Fatigue	0	2 (7.1%)	0	
Stomatitis (oral mucositis)	9 (32.1%)	0	2 (10.5%)	0
Peripheral neuropathy	6 (21.4%)	0	4 (21.1%)	0
Lacrimation (watery eyes)	6 (21.4%)	0	3 (15.8%)	0
Edema	3 (10.7%)	0	1 (5.3%)	0
Heart failure	0	0	0	0
Injection site reaction—Pain	5 (17.9%)	0	4 (21.1%)	0

* RBC: Red Blood Cells.

**Table 9 curroncol-32-00658-t009:** Clinical Characteristics and Treatment Outcomes in Patients with Metastatic HER2-Positive Breast Cancer Receiving PHESGO^®^ Therapy.

Variable	*n*	CR (*n*)	PR (*n*)	SD (*n*)	PD (*n*)	ORR (%)	DCR (%)
Metastatic site							
Lung/Pleura	3 *	0	1	1	1	33.3 (1/3)	66.6 (2/3)
Liver	3	0	3	0	0	100.0 (3/3)	100.0 (3/3)
Bone	3	0	3	0	0	100.0 (3/3)	100.0 (3/3)
Lymph nodes	5	0	3	0	2	60.0 (3/5)	60.0 (3/5)
Skin	3	0	1	0	2	33.3 (1/3)	33.3 (1/3)
Brain	1	0	0	1	0	0.0 (0/1)	100.0 (1/1)
Prior (neo)adjuvant HP exposure							
Yes	5 *	0	2	2	1	40.0 (2/5)	80.0 (4/5)
No	4	0	3	0	1	72.5 (3/4)	72.5 (3/4)
Co-administered taxane							
Docetaxel	5 *	0	3	1	1	60.0 (3/5)	80.0 (4/5)
Paclitaxel	1	0	0	0	1	0.0 (0/1)	0.0 (0/1)
None	3	0	2	1	0	66.6 (2/3)	100.0 (3/3)

* Number of cases excluding one unevaluated case. Note: CR, Complete Response; PR, Partial Response; SD, Stable Disease; PD, Progressive Disease; ORR (overall response rate) = CR + PR; DCR (disease control rate) = CR + PR + SD.

## Data Availability

The data presented in this study are available on request from the corresponding author.

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
