# Peer review of "Real-World Outcomes of Subcutaneous PHESGO^®^ in HER2-Positive Breast Cancer: Pathological Response, Sequencing, and Safety"

_curroncol, 2025, doi:10.3390/curroncol32120658_

Round 1
Reviewer 1 Report
Comments and Suggestions for Authors
Summary
This manuscript reports a retrospective study evaluating PHESGO® in 47 Japanese patients with HER2+ breast cancer across neoadjuvant, adjuvant, and metastatic settings. The study is the first real-world evidence in an Asian population, showing high neoadjuvant pCR rates, acceptable tolerability, and comparable efficacy to established intravenous regimens. The findings fill the gap left by FeDeriCa (phase 3 randomized trial) published in 2020 and may support broader use of subcutaneous dual HER2 blockade in routine practice. However, the study’s interpretation is limited by its small sample size and single-center design as stated by the authors. Overall, this manuscript is well-informed, except the metastatic outcomes section which would benefit from additional clarification to improve the readability and interpretability.
Below are my suggestions for improvement-
Major concerns:
- Introduction: The first paragraph currently has zero reference to support those statements. Statements like these that provide epidemiologic data, global statistics, and well-established disease characteristics should always be cited, even if they are “common knowledge” in oncology. Please provide proper references.
- Materials and Methods: This section is currently written in a single, continuous format, which is very challenging to follow and locate specific details. To enhance the readability, I strongly recommend the authors divide the current version into subsections with clear headings. This will help readers find information quickly and make the paper more professionally structured.
- Section 3.5: The authors reported that six patients had prior exposure to trastuzumab and pertuzumab in the (neo)adjuvant setting, but it is not specified how this subgroup responded to PHESGO® in the metastatic setting. Given that prior HER2-specific treatments can influence response and progression outcomes, summarizing the outcomes for these patients would improve interpretability and attract readers with clinical background. I strongly recommend including this information in the text or Table 4.
- In Section 3.5, the authors reported that PHESGO® was administered to 10 metastatic patients, but only 9 were evaluated without further explanation. The authors should clarify the reason for this discrepancy. Given the small cohort size, even one missing case can affect the interpretation of the response rate, and explicitly stating the reason is required for transparency and data integrity.
- Table 4: The authors provided an overview of metastatic sites, however, there is no information about how responses were distributed across these organ sites. The authors should provide a brief descriptive summary or this information in the table to enhance the interpretability.
Minor concerns:
- “NAC: Neoadjuvant chemotherapy” should be included in the list of abbreviations for readers who may not have a clinical background.
Author Response
Thank you for your feedback. I have compiled the points you raised into a file.

Reviewer 2 Report
Comments and Suggestions for Authors
This is a well written paper that presents the experience of one centre in Japan with the use of PHESGO a subcutaneous preparation of pertuzumab and trastuzumab with hyaluronidase in patients with HER2 positive breast cancer. While the number of patients is limited (N=47) and patients are from different clinical settings neoadjuvant, adjuvant and metastatic the authors did capture the experience of patients well assessing both clinical response, side effects and ease of use. The authors demonstrate that this combination of drugs is well tolerated clinically, is easier to administer and receive without compromising clinical efficiency. pCR rates in the neoadjuvant setting were high ( approximately 65%), and objective response rates in the metastatic setting were equivalent to current standard of care preparations. Side effects were well tolerated irrespective of patient age. A novel finding by this observational cohort implies that the sequencing of the drugs is important for outcome with PHESGO first regimens being superior in the neoadjuvant setting. Another important finding by the group is that current tumour biomarkers such as ER, PR, Ki67 and p53 do not help to identify a group of patients that will respond better or worse to the regimen in question.
In aggregate, this is a very well written article the only limitation of which is the sample size.
Author Response
Thank you for your feedback. I have included the response in the file.

Reviewer 3 Report
Comments and Suggestions for Authors
The objective of this study is to evaluate the clinical utility of PHESGO® in a real-world Japanese cohort across neoadjuvant, adjuvant, and metastatic settings. The manuscript is well written, with proper analysis, although with a rather small patient number. There is only one issue to be mentioned, that in the metastatic setting, PHESGO® was administered every three weeks, usually with docetaxel 60 mg/m² tri-weekly but not the standard 75 mg/m².
Author Response
Thank you for reviewing. I have noted the corrections in the file.
